


**A Comparative Analysis of Machine Learning Algorithms for Snowfall Prediction**
**Models in South Korea**
Moon-Soo Song[1], Hong-Sik Yun[2], Jae-Joon Lee[3], Sang-Guk Yum[4,*]
[1] Post-doctorate, Ph.D., Interdisciplinary Program in Crisis, Disaster and Risk Management, Sungkyunkwan
University, Suwon, 16419, Korea; sms0722@daum.net
[2] Post-doctorate, Ph.D., Interdisciplinary Program in Crisis, Disaster and Risk Management, Sungkyunkwan
University, Suwon, 16419, Korea; yoonhs@skku.edu
[3] Professor, Ph.D., School of Civil, Architectural Engineering & Landscape Architecture, Sungkyunkwan
University, Suwon, 16419, Korea; lunevocal1@naver.com
[4] Professor, Ph.D., Department of Civil Engineering, College of Engineering, Gangneung-Wonju National
University, Gangneung, 25457, Korea; skyeom0401@gwnu.ac.kr
*Correspondence to: Sang-Guk Yum (skyeom0401@gwnu.ac.kr)*
**Abstract**
Heavy snowfall is a natural disaster that causes extensive damage in South Korea. Therefore, it is
crucial to predict snowfall occurrence and establish countermeasures to reduce the damage caused by
heavy snowfall. In this study, the meteorological and geographic data of the past 30 years were collected,
and four machine learning algorithms were used: multiple linear regression (MLR), support vector
regression (SVR), random forest regressor (RFR), and eXtreme gradient boosting (XGB). Subsequently,
the performances of the machine learning algorithms were compared. Machine-learning algorithms
were selected as regression models to predict heavy snowfall. Additionally, grid search and five-fold
cross-validation techniques were used to improve learning performance. Model performance was
evaluated by comparing the observed and predicted data. It was observed that the RFR model accurately
predicted the occurrence of snowfall ($R^2$=0.64) compared with other models with various statistical
criteria. This result demonstrates the possibility of using the RFR model for heavy snowfall prediction.
The proposed study can aid the government, local governments, and public institutions in developing
strategies to respond to heavy snowfall in the fields of facilities, roads, and transportation.




**Keywords: snowfall prediction, machine learning, comparative analysis**




## 1. Introduction[1]

The 5th report of the IPCC stated that the abnormal climate observed worldwide is due to the rapid climate change caused by global warming (IPCC, 2014). Because of global warming, the ice in the Arctic region melts and subsequently evaporates to form a large number of clouds. This has increased the occurrence of heavy snowfall in the Northern Hemisphere, particularly in countries, such as Siberia. Heavy snowfall frequently occurs in the northern mid-latitudes (Krasting et al., 2013) and causes significant damage. In February 2021, shipments of COVID-19 vaccines to New York, USA, were suspended because of the heaviest snowfall in the past ten years. In January 2019, a snowstorm in Austria killed 11 people and isolated 12,000. In March 2018, heavy snowfall and cold waves in Europe killed 53 people. In December 2020, approximately 2,000 vehicles were isolated in Tokyo, Japan owing to heavy snowfall.

According to Article 3, No. 1 of the Framework Act on the Management of Disasters and SAFETY, in South Korea, heavy snowfall is classified as a major natural disaster. The damages caused by heavy

---

[1] **Abbreviations:**
Artificial neural network (ANN)
Automated synoptic observing system (ASOS)
Coefficient of determination ($R^2$)
Decision tree (DT)
eXtreme gradient boosting (XGB)
Gradient boosting machine (GBM)
Intergovernmental Panel on Climate Change (IPCC)
Korea Meteorological Administration (KMA)
Mean absolute error (MAE)
Ministry of the Interior and Safety (MOIS)
Multiple linear regression (MLR)
Random forest (RF)
Random forest regressor (RFR)
Representative concentration pathway (RCP)
Root mean square error (RMSE)
Snow ratio (SR)
Support vector machine (SVM)
Support vector regression (SVR)
Tolerance (TOL)
Variance inflation factor (VIF)



snowfall have been incurred nationwide in the safety fields of roads, logistics, transportation, and facilities.
According to the 'Disaster Annual Report 2019' published by the MOIS, which annually establishes and
publishes major statistics on the damage and recovery status of natural disasters, typhoon, heavy rainfall,
and heavy snowfall damage have accounted for approximately 53.85% ($1550 million) , 35.21% ($1014
million), and 6.47% ($186 million) of the total damage caused by natural disasters over the past 10 years
(2010–2019) (MOIS, 2020). Heavy snowfall has caused extensive damage in Korea, and studies on heavy
snow prediction and damage reduction are required.

55        Previous studies related to heavy snowfall prediction have been conducted primarily in

meteorology and climate. Recently, studies related to heavy-snow prediction have been conducted in
disaster management. The accumulated data on meteorological factors, such as temperature,
precipitation, and relative humidity, and geographic factors, such as altitude, latitude, and longitude,
were utilized to predict heavy snowfall. Research has been conducted using statistical and machine
learning techniques that can consider the nonlinear relationship of factors and SR, which is the ratio of
snowfall depth to the amount of liquid-equivalent precipitation (Byun et al., 2008). Because snow cover
occurs as a complex nonlinear combination of factors caused by meteorological and geographic
conditions, the nonlinear relationship between temperature, precipitation, relative humidity, and
geographic factors that affect snow cover should be considered (Park et al., 2016).

65        First, previous studies on snowfall prediction conducted in South Korea were reviewed. Kim et al.

(2013) collected temperature, precipitation, and snowfall data and developed a snowfall prediction
model using an ANN model and a multiple regression model. The ANN model exhibited better
performance than the multiple regression model. Park et al. (2014) developed a snowfall prediction
model by learning precipitation, minimum temperature, and maximum temperature as input variables
using an ANN and proposed a frequency analysis result to the RCP scenarios. In addition, a comparison
between the results of learning by individual weather stations with those of learning by the integrated
data demonstrated that the performance of the model trained by integrating the data of all points was
exceptional. Kim et al. (2014) used an ANN model to learn the temperature and precipitation data. In



addition, they calculated the probability of snow cover using the KMA-RegCM3 climate model and
climate change RCP scenario data provided by the KMA. Oh et al. (2020) conducted a study that
predicted the depth of snowfall by applying temperature and humidity changes and solar insolation
using multiple linear regression analysis.

78       Tabari et al. (2010) compared the predicted results derived using MLR, allowance ratio, and ANN,

using latitude, longitude, altitude, snow cover, and snow density as the input variables. A comparison
between the $R^2$ and RMSE values of the model determined that the MLR model yielded optimum results
with $R^2$ and RMSE values of 0.67 and 47.12, respectively. Liang et al. (2015) predicted snow depth in
Xinjiang, northern China, using data, such as visible and infrared surface reflectance, brightness, and
temperature using the SVM method. The performance of the SVM prediction model was evaluated by
using a correlation coefficient of 0.87. Hamidi et al. (2018) predicted monthly snowfall in Iran using
SVM, RF, and MLR methods. This study was conducted using time-series forecasting, and monthly
snowfall observation data were used as input variables. The performance of each model was evaluated
using RMSE and $R^2$ values, and it was observed that the SVM model exhibited exceptional performance
with an $R^2$ value of 0.95, which was applied for snowfall prediction in the area. Zhang et al. (2019)
performed snow-load predictions for mountainous regions. Eight factors, including average temperature,
relative humidity, wind speed, latitude, longitude, altitude, slope, and slope direction, were used as input
parameters for the MLR and RF models to predict snowfall. The coefficient of determination of the RF
model was 0.74, which was superior to that of the linear regression model. In addition, relative humidity,
temperature, and longitude were identified as the three crucial variables affecting snowfall. Hu et al.
(2021) derived a gridded predictive snowfall dataset using ANN, SVR, and RFR algorithms for five
regions in the northern hemisphere. The geographic location (latitude and longitude), topographic data
(altitude), and field observation data were used as input variables, and the RFR model exhibited the best
performance.

Recent studies have accurately predicted snowfall using various machine learning techniques. This

is because nonlinear activation functions (sigmoid and hyperbolic tangent) are used in machine learning

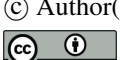



algorithms to evaluate the nonlinear relationship between weather factors. Learning results are
determined through trial and error (Tabari et al., 2010).

**2. Materials and methods**

**2.1 Study Area and data description**

The input variables used in previous studies were used to develop a snowfall prediction model. Table 1
shows that nine input variables were selected by dividing each factor into geographic (latitude, longitude,
and altitude of the ASOS) and meteorological factors (minimum temperature, maximum temperature,
average temperature, precipitation, relative humidity, and snowfall).

**Table 1. Geographic and meteorological factors for machine learning model training**

| Input Variables | | Output Variables |
|---|---|---|
| Geographic factors | Minimum temperature (°C), maximum temperature (°C), average temperature (°C), precipitation (mm), relative humidity (%) | Snowfall (cm) |
| Meteorological factors | Latitude (°), longitude (°), and altitude (m) | |

Meteorological data over the past 30 years (1991–2020) during the winter season (October to April) were
collected from 102 ASOS nationwide under the KMA. These factors included daily minimum temperature,
maximum temperature, average temperature, precipitation, and relative humidity. Figure 1 shows the
study area and ASOSs in South Korea.
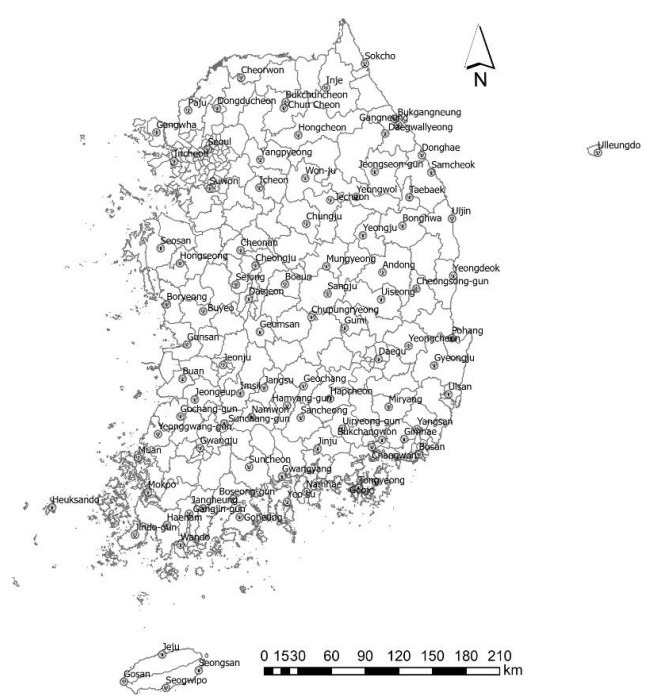


**Figure 1. Study area - ASOSs in South Korea**


Machine learning is difficult to perform when there are missing values in the dataset. Therefore, a
complete removal method was used to eliminate the datasets with missing independent variables.
Among the collected 945,748 daily datasets, 42,701 were selected after excluding non-snowy days and
datasets with missing values. In addition, a multicollinearity analysis was performed. Multicollinearity
is a problem that results in inaccurate analysis owing to the strong correlations between the independent
variables in the regression analysis. A general diagnostic index of multicollinearity states that a
multicollinearity problem occurs when the TOL is less than 0.1 or the VIF is greater than 10 (Ainiyah
et al., 2016). A high VIF indicates a high collinearity (Mallick et al., 2021). This study performed
multicollinearity analysis on meteorological factors (average temperature, minimum temperature,
maximum temperature, daily precipitation, and average relative humidity) and snowfall among




independent variables. Table 2 shows the results of the collinearity analysis. The VIF of the average
temperature was 21.738. After dimensionality reduction, the multicollinearity analysis was repeated by
excluding average temperature from the independent variable. The variance expansion coefficient of
the variables was ≤ 2, and it was verified that multicollinearity was absent.

**Table 2. Multicollinearity analysis**

|   | Input Variables | TOL | VIF |   | Input Variables | TOL | VIF |
|---|---|---|---|---|---|---|---|
| 1 | Average temperature (°C) | .046 | 21.738 | 2 | Average temperature (°C) | - | - |
|   | Minimum temperature (°C) | .104 | 9.585 |   | Minimum temperature (°C) | .533 | 1.877 |
|   | Maximum temperature (°C) | .149 | 6.689 |   | Maximum temperature (°C) | .561 | 1.783 |
|   | Precipitation (mm) | .816 | 1.226 |   | Precipitation (mm) | .816 | 1.226 |
|   | Relative humidity (%) | .849 | 1.178 |   | Relative humidity (%) | .849 | 1.178 |

**Output variables: snowfall (cm)**


The pre-processed datasets consisted of the final eight input variables, and four machine-learning
algorithms (MLR, SVR, RFR, and XGB) were trained. The snowfall prediction model was developed on
a Jupyter Notebook (64-bit Windows 10) using Python 3.7. The optimal hyperparameters for each
algorithm were selected and applied using a grid search technique during the learning process.
Additionally, the data were used for training using 5-fold cross-validation to improve accuracy and solve
the overfitting problem. The model performance was evaluated by comparing the snowfall estimated by
the trained model with the actual snowfall value measured at the observation station. The optimal model
was determined by comparing and verifying the accuracy of the models using MAE, RMSE, and $R^2$.
Figure 2 shows a graphical representation of the research workflow.




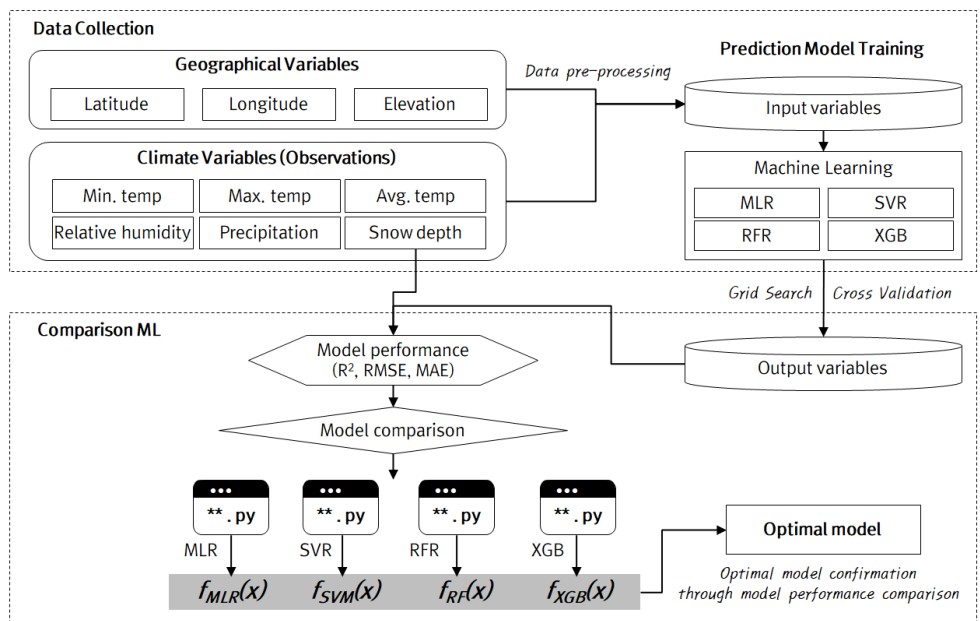


**Figure** 2. **Research workflow**


**2.2 MLR**
Linear regression is an extensively used regression analysis model, and it has been used by researchers
before the invention of artificial intelligence (Chaloulakou et al., 2003). This method derives the results
of independent and dependent variables using a one-dimensional linear predictive equation. The derived
equation when the cost function has a minimum value is defined as the optimal predictive model. The
least-squares method or gradient descent method is mainly used to determine the minimum value of the
loss function (Liu et al., 2021). Linear regression analysis refers to the estimation of a dependent
variable using a statistical method considering the independent variables $(X_1, X_2, \cdots, X_k)$ that are
expected to affect the dependent variable (Y) significantly. The linear regression model expresses the
relationship between the dependent and independent variables in linear form, as shown in Equation 1.


$$Y = a_0 + a_1X_1 + a_2X_2 + \ldots + a_kX_k,$$  Eq. 1


where $a_0$ represents the constant and $a_1$, $a_2$, and $\cdots$ $a_k$ are the regression coefficients of each independent
variable. A multiple regression analysis was performed for the independent variables (factors affecting
snowfall) in this study. Additionally, the variables were adjusted and analyzed after multicollinearity
analysis was performed.

**2.3 SVR**
SVM (Cortes & Vapnik, 1995) is a supervised machine learning algorithm used for classification
problems. The input variable is built into a high-dimensional functional space using a linear or nonlinear
kernel function depending on the relationship between the dependent and independent variables. A
linear model was developed in the feature space to maintain a balance between error minimization and
overfitting (Bansal et al., 2021). SVR is an extension of SVM that can be applied to classification
problems and prediction fields such as regression analysis (Bermolen & Rossi, 2009). SVR learns in a
direction that maximizes the distance between the separation hyperplane and support vector within a
threshold (Carrera & Kim, 2020).

**2.4 RFR**
The RF algorithm is a DT-based algorithm (Breiman, 2001). It is a model of an ensemble technique
developed by combining multiple DTs with different structures and performance. It functions by
outputting classification or average predictions (regression analysis) from multiple DTs that are
constructed during the training process. The RFR compensates for the bias introduced by a single DT
owing to the randomness. Therefore, it does not easily overfit and provides high accuracy and a fast
training speed (Babar et al., 2020). The RFR algorithm randomly selects data (bootstrapping) and learns
individually. Bagging is an abbreviation for bootstrap and aggregation, which is a concept that collects
models generated from each bootstrap sample. Aggregating refers to the merging of datasets formed by





bootstrapping, and a random subspace is applied to train the dataset. A random subspace is a process of
ensuring the independence of each basic algorithm. Determining the split point of the DT based on the
split function implies that learning is performed by randomly selecting a number of variables that are
less than the variables of the input data. In contrast to the DT algorithm, in which the error is transferred
at each intermediate node in RF, the error generated in the intermediate node of each tree is not
transmitted to the terminal node and converges to the limit value. This improves the predictive model's
performance by minimizing the correlation between individual trees (Ganguly et al., 2019).

**2.5 XGB**
XGB (Tianqi Chen & Guestrin, 2016) is known for its powerful performance, as demonstrated by recent
studies. In addition, they have been extensively used in various applications. XGB is an algorithm based
on GBM, a boosting model consisting of a series of basic regression trees using a sequential ensemble
technique (Zhu et al., 2021). This is a method of improving the error by sequentially repeating the
learning prediction for several weak learners and assigning weights when the predicted values differ
from the input data. The residual error of the model derived from Tree 1 was checked, and a predictive
model that reduced the residual error of Tree 1 was derived from Tree 2. Subsequently, the residuals in
Tree 2 are checked, and a predictive model that reduces the residuals in Tree 2 is derived using Tree 3.
This method derives a model from the final tree with small residuals as the final prediction model while
repeating this process (Zhu et al., 2021). Furthermore, XGB exhibits exceptional performance in
classification and regression problems. The weight of the hidden layer is not known in the case of
commonly used ANN-based algorithms. Therefore, the correlation between each variable and the
prediction model remains unknown. However, XGB has the advantage of being able to analyze the
feature importance of variables.

**2.6 Model performance**
Several criteria were used to evaluate the performance of the regression models. The accuracy of the




model was compared and verified using the MAE, MSE, RMSE, and $R^2$ values(Guo et al., 2021). The
MAE is the arithmetic mean of the absolute value of the difference between the measured and estimated
values. The MAE has high applicability if it has a value close to zero. The low MSE and RMSE values
demonstrate that the error of the estimation model was small. In this study, it was used to indicate the
suitability of the estimation of high snowfall (Hamidi et al., 2018). $R^2$ is used to measure the linear
relationship between the observed and estimated snowfall and has a value in the range 0–1. An $R^2$ value
close to 1 indicates optimum model applicability. The MAE, MSE, RMSE, and $R^2$ were calculated
using Equations 2, 3, 4, and 5, respectively.

$MAE = \frac{1}{m}\sum_{i=1}^{m}|X_i - Y_i|,$   Eq. 2
$MSE = \frac{1}{m}\sum_{i=1}^{m}(X_i - Y_i)^2,$   Eq. 3
$RMSE = \sqrt{\frac{1}{m}\sum_{i=1}^{m}(X_i - Y_i)^2},$   Eq. 4
$R^2 = 1 - \frac{\sum_{i=1}^{m}(X_i - Y_i)^2}{\sum_{i=1}^{m}(\bar{Y} - Y_i)^2},$   Eq. 5
where $X_i$ is the predicted $i_{th}$ value and $Y_i$ is the actual $i_{th}$ value. The regression method predicts
the $X_i$ element for the corresponding $Y_i$ element in the observation dataset (Chicco et al., 2021).





**2.7. Grid search and K-fold cross-validation**
The optimization of a regression model using machine learning refers to the estimation of a
hyperparameter that minimizes a predefined loss function in the training data(Luo, 2016). This study
applied the grid search and k-fold cross-validation methods to select the optimal hyperparameter. The grid
search depicted in Figure 3 was used to select the optimal parameters for each model. The range of each
parameter was set, the accuracy of the model generated according to the combinations was measured, and
the optimal parameter that provided the highest accuracy was selected (Claesen & De Moor, 2015). In the
case of the k-fold cross-validation method, as shown in Figure 4, the datasets were k equalized into sets
of the same size. The k-1 among the divided datasets was used as the training data, and the remaining
dataset was used as the testing data. This method was used to verify the performance of the model. In this
study, 5-fold cross-validation was applied (Vabalas et al., 2019).

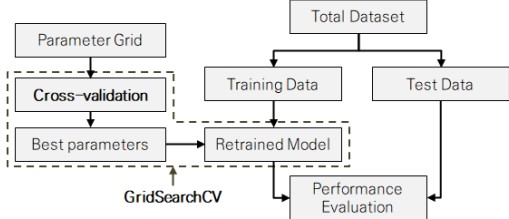


**Figure 3. Hyperparameter tuning using GridSearch**

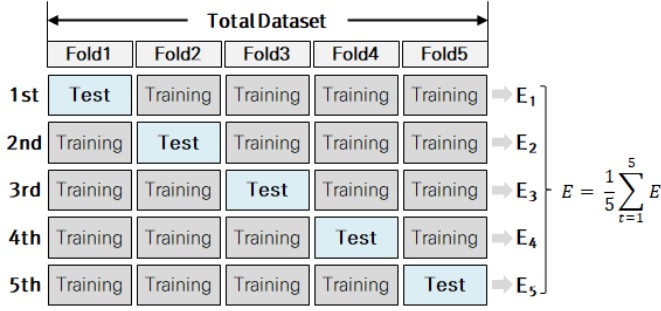


**Figure 4. 5-fold cross-validation**





**3. Result**
The optimum hyperparameter results of each machine-learning algorithm were derived through grid
search and k-fold cross-validation (Table 3).

**Table 3. Results of hyperparameter tuning**

| Models | Evaluated Hyperparameters | | Hyperparameters |
|---|---|---|---|
| SVR | Kernel | Linear, Polynomial, Sigmoid, RBF | RBF |
| | Cost | 0.01, 0.1, 1, 10, 100 | 1 |
| | $\gamma$ | 0.01, 0.1, 1, 10, 100 | 1 |
| RFR | max_features | 4, 8, 10, 12, 14, 16, 18, 20 | 4 |
| | n_estimators | 10~1000 | 100 |
| | max_depth | 4, 8, 10, 12 | 10 |
| XGB | max_features | 4, 8, 10, 12, 14, 16, 18, 20 | 4 |
| | n_estimators | 10~1000 | 20 |
| | max_depth | 4, 8, 10, 12 | 6 |


The applicability of $f_{MLR}(x)$, $f_{SVR}(x)$, $f_{RFR}(x)$, and $f_{XGB}(x)$, which were the optimal models for each
algorithm, was evaluated using hyperparameters. The RFR model exhibited MAE, MSE, RMSE, and $R^2$
values of 1.65, 11.68, 3.35, and 0.64, respectively, using performance evaluation criteria. Additionally, it
exhibited a higher prediction accuracy than the three models (MLR, SVR, and XGB models). The XGB
model exhibited a similar performance to the RFR model because it was close to the evaluation standard
value obtained based on the RF model. In the case of snowfall prediction, it was determined that ensemble
models, such as RFR and XGB, demonstrated better performance than single regression models such as
MLR and SVR.

**Table 4. Comparative statistics of prediction models**

| Criteria / Models | MAE | MSE | RMSE | $R^2$ |
|---|---|---|---|---|
| MLR | 2.32 | 18.20 | 4.22 | 0.45 |
| SVR | 1.73 | 15.91 | 3.91 | 0.53 |



| | | | | |
|---|---|---|---|---|
| **RFR** | 1.65 | 11.68 | 3.35 | 0.64 |
| **XGBoost** | 1.64 | 12.31 | 3.44 | 0.62 |






The snowfall prediction estimates obtained using the MLR, SVR, RFR, and XGB models and the
corresponding observed snowfall values are shown in Figures 5 through scatter plots. It was observed
that the snowfall simulation of the RFR and XGB models exhibited better performance compared with
that of the other two models. The RFR and XGB models accurately evaluated the nonlinear relationship
between the predictor and independent variables using a coefficient of determination. The MLR and
SVR models partially interpreted the variance in snowfall. In the case of field observation data, there is
a lack of datasets for high snowfall and there are a lot of datasets for low snowfall. The imbalance of
datasets was analyzed as a result of underestimating the MLR and SVR models(Park et al., 2021).
Finally, a comparison between the statistical criteria of the four models demonstrated that the RFR was
the optimum model for predicting snowfall. The predictive performance of the RFR model was
exceptional because it was not necessary to assume a correlation between the dependent and
independent variables in this model. In addition, it is less sensitive to datasets with inappropriate error
distributions (Zhang et al., 2019).
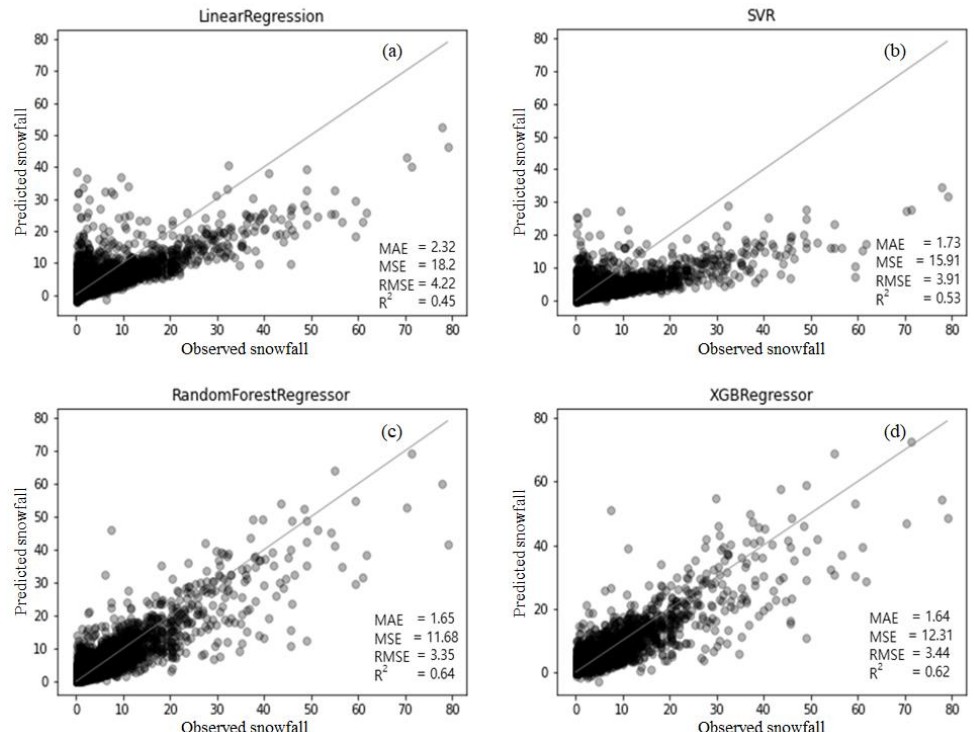

**Figure 5. Correlation of observed and predicted snowfall results from (a) MLR, (b) SVR, (c) RFR, and (d) XGB**

## 4. Discussion and Conclusions

In this study, the occurrence of snowfall over the past 30 years in Korea was investigated, and machine-learning algorithms were used to predict heavy snowfall. The optimal snowfall prediction model was selected to establish response strategies for heavy snowfall.

The snowfall prediction model was developed according to the following steps. Independent variables were selected by analyzing previous studies, and data collection was performed by considering the meteorological and geographic factors collected through the ASOS. Data pre-processing was performed, and the pre-processed data were learned using MLR, SVR, RFR, and XGB machine learning



algorithms. A machine learning algorithm was selected as the regression model for prediction purposes.
Grid search and k-fold cross-validation were used to improve learning performance. It was observed
that the predictive model using the RFR algorithm had the best performance based on a comparison
between the observed and predicted data. In addition, it was observed that the performance of the
ensemble models (RFR and XGB) was better than that of the single regression models (MLR and SVM).
Snowfall prediction is a nonlinear process in which precipitation, temperature, relative humidity, and
geographic variables are correlated. Additionally, the prediction results may vary depending on the
regional research scope and characteristics of the input variable data used for model development. The
meteorological factors were provided in the form of daily data when they were used as input variables.
Because the daily average observation data were used as input data for the meteorological factor, rather
than the weather data when the actual heavy snowfall occurred, the performance of the prediction model
was relatively low. In the future, the proposed model can be used as an estimation model to obtain the
distribution of the predicted snowfall in South Korea using the RCP climate change scenario.
Additionally, the model can aid in establishing response strategies for heavy snowfall disasters in road
facilities and transportation sectors by providing long-term prediction (~2100 years) data for heavy
snowfall. In particular, when predicting future snowfall using climate change RCP scenario data, it is
difficult to improve the predictive power of the model considering the uncertainty of the scenario.
Therefore, it is crucial to continuously develop and verify predictive models (Park et al., 2016).

**Author's contribution**
Moon-Soo Song: Conceptualization, Methodology, Data curation, Investigation, Writing – original,
review & editing.
Hong-Sik Yun: Conceptualization, Methodology, Funding
Jae-Joon Lee: Methodology, Investigation, Writing review & editing
Sang-Guk Yum: Methodology, Project administration, Validation, Supervision, Writing - review &
editing.




**Data Availability**

The data presented in this research are available from the corresponding author by reasonable request.

**Declaration of interests**

The authors declare that they have no known competing financial interests or personal relationships that
could have appeared to influence the work reported in this paper.

**Acknowledgment**

This paper was supported by research funds for newly appointed professors of Gangneung-Wonju
National University in 2021

**Funding**

This research was supported by a grant (2021-MOIS35-003) of 'Policy-linked Technology
Development Program on Natural Disaster Prevention and Mitigation' funded by the Ministry of Interior
and Safety (MOIS, Korea).

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
