# Peer review of "A Comparative Analysis of Machine Learning Algorithms for Snowfall Prediction"

_Natural Hazards and Earth System Sciences, 2022_

## Author Comment (AC2)

**Manuscript Number: NHESS-2022-118**

We sincerely thank the Reviewer 2 for the time in effort on reviewing our manuscript with many insightful comments. We believe that we have addressed each of the comments carefully and properly, while improving the quality of paper significantly. We hope that the changes listed below are acceptable for publication. In addition, we have made significant changes in all the relevant main text body, which could be aligned well with our responses to the comments in the revised manuscript. The changes made in the revised manuscript are highlighted in blue to facilitate their identification.
* * *
**General Comment:** This manuscript compared the skill of four machine learning algorithms, including multiple linear regression (MLR), support vector regression (SVR), random forest regressor (RFR), and eXtreme gradient boosting (XGB) for snowfall estimation in South Korea. Meteorological data (minimum temperature, maximum temperature, precipitation, and relative humidity) from 1991–2020 during the winter season (October to April) collected from the automated synoptic observing system, and geographic data (latitude, longitude, and altitude) were used as the input variables and the measured snow depth was used as the output variable for machine learning model training. The results indicate the RFR performs the best among the four machine learning algorithms with an $R^2$ of 0.64.

The work is interesting, however, the main drawback of this work is that it is too basic and simple. A great deal of similar works have been carried out in previous studies, and some of them have been summarized by the authors (Line 55-97). In the introduction, the authors only mentioned such previous works, but did not point out the problem which remains to be solved in the current work (i.e., the motivation of this study). In other words, if the paper is only a simple imitation of previous studies, it is not innovative.
* * *
Response: First, thank you for positively viewing our research ideas, with very insightful comments listed below. This study is a study to evaluate the applicability of various regression machine learning methods for predicting heavy snowfall in South Korea. Among the machine learning algorithms used in previous studies, models with good regression results were selected and applied to this study.

The snowfall prediction model of this study used four machine learning algorithms (MLR, SVM, RF, XGB) to learn the meteorological factors and geographic factors collected through the 102 ASOSs. This model will be used for GIS-based predicted snowfall distribution according to future RCP climate change scenarios. The four machine learning algorithms were selected as regression models for the purpose of prediction, not for identifying the cause of the heavy snowfall, and grid search and k-fold cross-validation techniques were used to improve learning performance. In addition, it is meaningful that geographic factors (latitude, longitude, altitude) as input

data that were not considered in the study of snowfall prediction in Korea.

We have re-written the Chapter 1 to clearly present this research's superiority such as contributions and novelty, as follows:

*Lines 95-110*

Prediction of snowfall in previous studies is a non-linear process in which precipitation, temperature, relative humidity, and geographic variables are variously related. Various machine learning techniques that can take this non-linear process have shown good results in predicting the amount of snowfall. This is because nonlinear activation functions (Sigmoid and Tanh) are used in machine learning algorithms to explain the nonlinear relationship between input factors(Tabari et al., 2010). However, the prediction results may vary greatly depending on the regional research scope and the characteristics of the input variable data used for model development. In this study, South Korea as the study area, input variables not applied in existing domestic studies from previous studies were synthesized and heavy snowfall prediction was performed using an excellent machine learning algorithm. In addition, the predictive model derived through this study can be used for GIS-based predicted snowfall distribution according to future RCP climate change scenarios and heavy snowfall disaster management.
* * *
**Comment 1**

Line 41-45: add references.
* * *
Response: We added 4 more references of news article webpages for the heavy snowfall events.

1. Associated Press. (2018). *Waves of Winter Storms Kill at Least 16 in Europe*. The Weather Channel.
2. Deutsche Welle. (2020). *Japan: Heavy snowfall leaves thousands stranded*. Deutsche Welle.
3. France24. (2021). *Huge snowstorm blankets US East Coast, halting travel and vaccinations*.
4. United Press International. (2019). *Major winter storm kills 4 in Germany and Austria*. Gephardtdaily.

*Line 36-42*

In February 2021, shipments of COVID-19 vaccines to New York, USA, were suspended because of the heaviest snowfall in the past ten years(France24, 2021). In January 2019, a snowstorm in Austria killed 4 people and isolated 12,000 tourists(United Press International, 2019). In March 2018, heavy snowfall and cold waves in Europe killed 16 people and More than 350 flights were canceled(Associated

Press, 2018). In December 2020, Around 1,000 cars have been stranded and about 10,000 households cut off power in a snowstorm in Japan(Deutsche Welle, 2020).
* * *
**Comment 2**

Line 81: where is the reference of "Liang et al. (2015)"?
* * *
Response: We added the reference of "Liang et al. (2015)" in References, the last section of the manuscript.

*Line 385-387*

Liang, J., Liu, X., Huang, K., Li, X., Shi, X., Chen, Y., & Li, J. (2015). Improved snow depth retrieval by integrating microwave brightness temperature and visible/infrared reflectance. Remote Sensing of Environment, 156(February), 500–509. https://doi.org/10.1016/j.rse.2014.10.016
* * *
**Comment 3**

Section 2.1: only meteorological data were used in the study. Due to the limited spatial coverage of the stations, why the authors did not consider other large-scale data such as remote sensing data or model (reanalysis) based data?
* * *
Response: In this study, in addition to meteorological data, which is the actual observation data from the ASOSs, latitude, longitude, and altitude information, which are geographical data of the ASOSs, were used. To improve the prediction accuracy of the machine learning model, real meteorological data with high spatiotemporal resolution were used. The remote sensing data was not considered because of spatiotemporal resolution issues, the issue of paying money for data collection, and issues that could not be used in bad weather conditions.

We hope that this is acceptable and reasonable. Thank you.
* * *
**Comment 4**

Fig. 1: this figure lacks longitude and latitude information. Moreover, its quality can be improved, e.g., you can use the legend information to represent the stations but do not need to list all the station names.
* * *
Response: As requested, we have added longitude and latitude information at the border of figure. Also, we confirmed that the names of all the stations were unnecessary and removed them.

*Line 128 (Figure 1.)*

[Figure]

**Figure 1. Study area – 102 ASOSs in South Korea**
* * *
**Comment 5**

Line 129: where is the reference of "Ainiyah et al., 2016"?

Response: We added the reference of "Ainiyah et al. (2016)" in References, the last section of the manuscript.

*Line 324-327*

Ainiyah, N., Deliar, A., & Virtriana, R. (2016). The classical assumption test to driving factors of land cover change in the development region of northern part of west Java. International Archives of the Photogrammetry, Remote Sensing and Spatial Information Sciences - ISPRS Archives, 41(July), 205–210. https://doi.org/10.5194/isprsarchives-XLI-B6-205-2016
* * *
**Comment 6**

Line 130: where is the reference of "Mallick et al., 2021"?

Response: We added the reference of "Mallick et al.(2021)" in References, the last section of the manuscript.

*Line 395-398*

Mallick, J., Alqadhi, S., Talukdar, S., Alsubih, M., Ahmed, M., Khan, R. A., Kahla, N. Ben, & Abutayeh, S. M. (2021). Risk assessment of resources exposed to rainfall induced landslide with the development of gis and rs based ensemble metaheuristic machine learning algorithms. Sustainability (Switzerland), 13(2), 1–30. https://doi.org/10.3390/su13020457
* * *
**Comment 7**

Line 140: should be seven inputs and one output?
* * *
Response: Yes, this should be "seven inputs and one output". We fixed in it in the revised manuscript.

*Line 148-149*

The pre-processed datasets consisted of the final seven inputs and one output variables, and four machine-learning algorithms (MLR, SVR, RFR, and XGB) were trained.
* * *
**Comment 8**

Fig. 2: isn't the average temperature excluded due to the high collinearity issue?
* * *
Response: Yes, average temperature was excluded because of high multicollinearity. However, the average temperature in the box of 'climate variables' was maintained because it was before the data preprocessing process. To avoid confusion for the readers, 7 inputs are displayed in the box of 'Input variables' and the Figure2 has been modified.

*Line 158 (Figure 2.)*

[Figure]

Figure 1. Research workflow
* * *
**Comment 9**

Line 153: it is better to add a section entitled "2.2 Machine Learning Methods" before "2.2 MLR". Moreover, there are numerous machine learning methods, why did you select the four methods?
* * *
Response: According to your kind review, the structure of chapter 2 has been revised and reflected in the revised version(2.2 Machin learning methods/2.2.1 MLR, 2.2.2 SVR, 2.2.3 RFR, 2.24 XGB). In the selection process of the machine learning methods, first, the regression model was suitable for predicting snowfall through the analysis of previous studies. Among the machine learning algorithms that support the regression, the algorithms with good results was selected in the preceding study like SVR, RFR and XGB. In the case of MLR, as mentioned in the manuscript, it was selected for comparison with the other three regression models.
* * *
**Comment 10**

Line 198: delete "Tianqi".
* * *
Response: Yes, this should be "(Chen & Guestrin, 2016)". We fixed in it in the revised manuscript.

*Line 208*

XGB (Chen & Guestrin, 2016) is known for its powerful performance, as demonstrated by recent studies.

**Comment 11**

Line 215: MSE and RMSE play the same role in the evaluation. You can only preserve RMSE.

Response: Yes, we fixed in it in the revised manuscript by deleting MSE and keeping only RMSE.

*Line 224-225*

The accuracy of the model was compared and verified using the MAE, RMSE, and R2 values(Guo et al., 2021)

**Comment 12**

Line 255-256: add unit for MAE, MSE, and RMSE.

Response: We added "cm" unit for MAE and RMSE.

*Line 263-264*

The RFR model exhibited MAE, RMSE, and R2 values of 1.65cm, 3.35cm, and 0.64, respectively, using performance evaluation criteria.

**Comment 13**

Table 4: add unit for MAE, MSE, and RMSE.

Response: We added "cm" unit for MAE and RMSE.

*Line 271 (Table 4.)*

Table 1. Comparative statistics of prediction models

| Criteria / Models | MAE(cm) | RMSE(cm) | $R^2$ |
|---|---|---|---|
| MLR | 2.32 | 4.22 | 0.45 |
| SVR | 1.73 | 3.91 | 0.53 |
| RFR | 1.65 | 3.35 | 0.64 |
| XGBoost | 1.64 | 3.44 | 0.62 |

**Comment 14**

Fig. 5: add unit for snowfall.

Response: We added "cm" unit for snowfall.

*Line 286(Figure 5.)*

[Figure]

Figure 2. Correlation of observed and predicted snowfall results from (a) MLR, (b) SVR, (c) RFR, and (d) XGB

---

## Author Comment (AC3)

**Manuscript Number: NHESS-2022-118**

We sincerely thank the Reviewer 2 for the time in effort on reviewing our manuscript with many insightful comments. We believe that we have addressed each of the comments carefully and properly, while improving the quality of paper significantly. We hope that the changes listed below are acceptable for publication. In addition, we have made significant changes in all the relevant main text body, which could be aligned well with our responses to the comments in the revised manuscript. The changes made in the revised manuscript are highlighted in blue to facilitate their identification.
* * *
**General Comment:** The authors compared four machine learning algorithms and derived an optimal model to predict heavy snowfall; however, the method and analysis process used are general and lack novelty.
* * *
Response: First, thank you for positively viewing our research ideas, with very insightful comments listed below.

The snowfall prediction model of this study used four machine learning algorithms (MLR, SVM, RF, XGB) to learn the weather factors and geographic factors collected through the 102 ASOSs. This model will be used for GIS-based predicted snowfall distribution according to future RCP climate change scenarios. The four machine learning algorithms were selected as regression models for the purpose of prediction, not for identifying the cause of the heavy snowfall, and grid search and k-fold cross-validation techniques were used to improve learning performance. In addition, it is meaningful that geographic factors (latitude, longitude, altitude) as input data that were not considered in the study of snowfall prediction in Korea.

We hope that this is acceptable and reasonable. Thank you.
* * *
**Comment 1**

In Chapter 1. Introduction, the authors only mentioned previous research, but did not describe this research's superiority such as contributions and novelty of the research.
* * *
Response: The key of this study is to secure the accuracy of heavy snowfall predicting for the vulnerability assessment to future heavy snow disasters. Therefore, in this study, four algorithms with good performance were selected and used through analysis of previous studies. In addition, the most used input factors were synthesized and classified into meteorological factors and geographic factors and used as input data. South Korea was designated as the case study area, and observation data from an actual weather station was used, not remote sensing data, considering the spatiotemporal resolution. As mentioned in 'Chapter 4 Discussion and Conclusion' of the manuscript, the predictive model derived through this study can be used for GIS-based predicted

snowfall distribution according to future RCP climate change scenarios.

We have re-written the Chapter 1 to clearly present this research's superiority such as contributions and novelty, as follows:

*Lines 95-110*

Prediction of snowfall in previous studies is a non-linear process in which precipitation, temperature, relative humidity, and geographic variables are variously related. Various machine learning techniques that can take this non-linear process have shown good results in predicting the amount of snowfall. This is because nonlinear activation functions (Sigmoid and Tanh) are used in machine learning algorithms to explain the nonlinear relationship between input factors(Tabari et al., 2010). However, the prediction results may vary greatly depending on the regional research scope and the characteristics of the input variable data used for model development. In this study, South Korea as the study area, input variables not applied in existing domestic studies from previous studies were synthesized and heavy snowfall prediction was performed using an excellent machine learning algorithm. In addition, the predictive model derived through this study can be used for GIS-based predicted snowfall distribution according to future RCP climate change scenarios and heavy snowfall disaster management.
* * *
**Comment 2**

In Chapter 2. Materials and methods, it should have mentioned why the four machine learning algorithms (MLR, SVR, RFR, and XGB) were chosen among the other various machine learning algorithms.
* * *
Response: This study is to evaluate the applicability of various regression machine learning methods for predicting heavy snowfall in South Korea. Many machine learning techniques for regression and prediction have been proposed in the literature. Among the machine learning algorithms used in previous studies, models with good regression results were selected and applied to this study. In this paper, we focused on three leading approaches that have proven to be most effective in a wide variety of applications, i.e., support vector regression (SVR), random forest regressor (RFR) and extreme gradient boosting(XGB).

SVR most widely used in various prediction fields and have also been applied to prediction of snowfall and frost occurrence. RFR shows the best performance in similar previous studies and has the advantage of being able to check the importance scores of various variables. XGB is more recent, and its application for prediction is emerging. In the case of MLR, as mentioned in the manuscript, it was selected for comparison with the other three regression models.

The table below shows machine learning algorithms and compares the performance of

each model through the coefficient of determination used in previous studies. Random Forest and SVM show high performance.

| References | Machine learning algorithms | Performance($R^2$) |
|---|---|---|
| Tabari et al., 2013 | MLR, ANN, NNGA | NNGA |
| Park et al., 2014 | ANN, MLR | ANN> MLR |
| Ahn et al., 2015 | SR | - |
| Park et al., 2016 | MLR | - |
| Oh et al., 2017 | MLR | - |
| Hamidi et al., 2018 | RF, SVM, MARS | RF>SVM>MARS |
| Ki et al., 2018 | RF, GLM, NNET | RF>GLM≈NNET |
| Zhang et al., 2019 | RF, MLR, RNN | RF>RNN>MLR |
| Hu et al., 2021 | ANN, SVM, RF | RF>SVM>ANN |

We mentioned the Chapter 1 to explain why the four machine learning algorithms (MLR, SVR, RFR, and XGB) were chosen among the other various machine learning algorithms in this research, as follows:

*Line 95-98*

Among the machine learning algorithms used in previous studies, models with good regression results were selected and applied to this study. In this paper, we focused on three leading approaches that have proven to be most effective in a wide variety of applications, i.e., support vector regression (SVR), random forest regressor(RFR) and extreme gradient boosting(XGB) (Bedi et al., 2020).
* * *
**Comment 3**

In Chapter 4. Discussion and Conclusions, specific discussions on applicability of the optimal model, which is connected with RCP scenario and heavy snowfall disaster management, should be presented. Furthermore, there were a few sentences to confuse readers (for example, lines 269-272). The authors need to clarify those sentences in the manuscript and proofread the manuscript before submission.
* * *
Response: We are going to modify the 'Chapter 4. Discussion and Conclusions' to add

the specific explanation of RCP scenario and heavy snowfall disaster management.

*Line 309-313*

In the future, the proposed model can be used as an estimation model to obtain the distribution of the predicted daily snowfall in South Korea using the RCP climate change scenario data. Predicted daily snowfall data with RCP scenario can aid in establishing response and management strategies for heavy snowfall disasters in road facilities and transportation sectors by providing long-term snowfall prediction data until 2100.

Also, we fix the sentences (lines 269-272 / 274-277(revised version)) to clarify the results of the MLR and SVR models underestimating snowfall.

*Line 278-281*

The MLR and SVR models partially interpreted the variance in snowfall. In the case of field observation data, there is a lack of datasets for high snowfall and there are a lot of datasets for low snowfall. The imbalance of datasets was analyzed as a result of underestimating the MLR and SVR models(Park et al., 2021).

We have undergone proofreading twice for this thesis, but we will have it proofread once more before final submission.
* * *
**Comment 4**

Furthermore, there are additional explanation of the a, b, c, and d of figure 5 should be added to make clear to the readers.
* * *
Response: We added the explanation of (a) MLR, (b) SVR, (c) RFR, and (d) XGB.

*Line 273-275*

The predicted snowfall values obtained using the MLR, SVR, RFR, and XGB models and the observed snowfall values are shown in Figures 5 through scatter plots of (a) MLR, (b) SVR, (c) RFR, and (d) XGB

[Figure]

**Comment 5**

Please add more relevant literature review with up-to-date.

Response: We added 7 more relevant literate review.

1. Ainiyah, N., Deliar, A., & Virtriana, R. (2016). The classical assumption test to driving factors of land cover change in the development region of northern part of west Java.
2. Associated Press. (2018). *Waves of Winter Storms Kill at Least 16 in Europe*. The Weather Channel.
3. Bedi, S., Samal, A., Ray, C., & Snow, D. (2020). Comparative evaluation of machine learning models for groundwater quality assessment.
4. Deutsche Welle. (2020). *Japan: Heavy snowfall leaves thousands stranded*. Deutsche Welle.
5. France24. (2021). *Huge snowstorm blankets US East Coast, halting travel and vaccinations*.
6. Mallick, J., Alqadhi, S., Talukdar, S., Alsubih, M., Ahmed, M., Khan, R. A., Kahla, N. Ben, & Abutayeh, S. M. (2021). Risk assessment of resources exposed to rainfall induced landslide with the development of gis and rs based ensemble metaheuristic machine learning algorithms.
7. United Press International. (2019). *Major winter storm kills 4 in Germany and Austria*. Gephardtdaily.